

# Association between C-Maf-inducing protein gene rs2287112 polymorphism and schizophrenia

Yingli Fu[1,2], Xiaojun Ren[3], Wei Bai[4], Qiong Yu[2], Yaoyao Sun[2], Yaqin Yu[2] and Na Zhou[5]

[1] Division of Clinical Research, First Hospital of Jilin University, Changchun, Jilin, China
[2] Department of Epidemiology and Biostatistics, Jilin University, School of Public Health, Changchun, Jilin, China
[3] Department of Radiation Oncology, The Second Hospital of Jilin University, Changchun, Jilin, China
[4] Center for Cognition and Brain Sciences, University of Macau, Macao SAR, China
[5] State Key Laboratory of Quality Research in Chinese Medicine and School of Pharmacy, Macau University of Science and Technology, Macao SAR, China

## ABSTRACT

**Background:** Schizophrenia is a severely multifactorial neuropsychiatric disorder, and the majority of cases are due to genetic variations. In this study, we evaluated the genetic association between the C-Maf-inducing protein (*CMIP*) gene and schizophrenia in the Han Chinese population.

**Methods:** In this case-control study, 761 schizophrenia patients and 775 healthy controls were recruited. Tag single-nucleotide polymorphisms (SNPs; rs12925980, rs2287112, rs3751859 and rs77700579) from the *CMIP* gene were genotyped *via* matrix-assisted laser desorption/ionization time of flight mass spectrometry. We used logistic regression to estimate the associations between the genotypes/alleles of each SNP and schizophrenia in males and females, respectively. The in-depth link between *CMIP* and schizophrenia was explored through linkage disequilibrium (LD) and further haplotype analyses. False discovery rate correction was utilized to control for Type I errors caused by multiple comparisons.

**Results:** There was a significant difference in rs287112 allele frequencies between female schizophrenia patients and healthy controls after adjusting for multiple comparisons ($\chi^2 = 12.296$, $P_{adj} = 0.008$). Females carrying minor allele G had 4.445 times higher risk of schizophrenia compared with people who carried the T allele ($OR = 4.445$, 95% CI [1.788–11.046]). Linkage-disequilibrium was not observed in the subjects, and people with haplotype TTGT of rs12925980–rs2287112–rs3751859–rs77700579 had a lower risk of schizophrenia ($OR = 0.42$, 95% CI [0.19–0.94]) when compared with CTGA haplotypes. However, the association did not survive false discovery rate correction.

**Conclusion:** This study identified a potential *CMIP* variant that may confer schizophrenia risk in the female Han Chinese population.

Corresponding authors
Yaqin Yu, yuyaqin5540@163.com
Na Zhou, nzhou@must.edu.mo

## INTRODUCTION

Schizophrenia (SCZ) is a severely multifactorial neuropsychiatric disorder that affects almost 1% of adults around the world. A recent study found that the lifetime prevalence of SCZ patients in China was 0.6% (*Huang et al., 2019*). SCZ has devastating impacts on patients' and their families' quality of life. It also has an enormous financial cost. SCZ is a prototypical multifactorial disorder caused by both genetic and environmental factors. Genetic factors play a major role in SCZ etiology (*Owen, Sawa & Mortensen, 2016*) and genetic variations in chromosome 16 are associated with a variety of neuropsychiatric disorders. Some rare, common, and copy number variants on chromosome 16p have been found to be associated with SCZ (*Chang et al., 2017*; *Giaroli et al., 2014*). Regions on chromosome 16q, highly specific to a single psychometric measure, are also associated with neuropsychiatric disorders. Previous studies found that regions on chromosome 16q may increase susceptibility to SCZ (*Lewis et al., 2003*), bipolar disorder (*Lewis et al., 2003*), and autism (*Wassink et al., 2008*). Furthermore, large-scale genome-wide association studies (GWAS) conducted by *Bigdeli et al. (2020)* and *Pardiñas et al. (2018)* respectively showed two (rs34753377 and rs6500603) and three (rs17465671, rs12447542 and rs2161711) single-nucleotide polymorphisms (SNPs), located on chromosome 16 that were associated with SCZ.

C-Maf-inducing protein (*CMIP*) is an important gene located on 16q23 that is mainly expressed in human brains, encodes an 86-kDa protein 7-9, and plays a role in the T-cell signaling pathway (*Liu et al., 2015*). *CMIP* contributes to several biological pathways and is involved in various diseases such as glioma, gastric cancer, kidney disease, and dyslipidemia (*Li et al., 2019*; *Mo et al., 2018*; *Wang & Wu, 2017*; *Zhang et al., 2017*), as well as major depressive disorder, syndromic autism spectrum disorders, and specific language impairments (*Eicher & Gruen, 2015*; *Gedik, 2017*; *Luo et al., 2017*; *Wang et al., 2015*). However, no studies have documented the relationship between *CMIP* and SCZ.

Based on chromosome 16's biological function and previous studies on *CMIP*, we hypothesized that *CMIP* may have a relationship with SCZ. Additionally, gender-specific associations between gene SNPs (*i.e.*, *RELN, GABRB3* and *MTHFR*) and SCZ have been found in several other studies (*Sozuguzel & Sazci, 2019*; *Wan et al., 2019*; *Liu et al., 2018*). We conducted a genetic association study stratified by gender to examine the association between tag SNPs of the *CMIP* gene and SCZ in the Han Chinese population.

## MATERIALS & METHODS

### Study sample

A total of 761 SCZ patients and 775 healthy controls without any personal or family history of mental illness were enrolled in this study. More details of the data collection are described in a previous paper (*Fu et al., 2020*). All subjects were recruited after providing written informed consent. The study was performed in accordance with the protocols approved by the Ethics Committee of Jilin University, China (2014-05-01).

**Table 1 Primers for polymerase chain reaction.**

| SNP | Primer sequence (5′–3′) | |
| --- | --- | --- |
| | **Forward** | **Reverse** |
| rs2287112 | ACGTTGGATGATCAGCAAGAGCCTCAAACC | ACGTTGGATGTGGTTGCTGGTCTGCTTTTC |
| rs77700579 | ACGTTGGATGAGGATAGTGAGCACTTACCC | ACGTTGGATGGACAATGACAGCACCACCTC |
| rs3751859 | ACGTTGGATGTTTCCACCAGTGCTCAGGG | ACGTTGGATGGTTCTCCAGGTTCAAATGTC |
| rs12925980 | ACGTTGGATGCCCTTCCCCCATTGATACTC | ACGTTGGATGCACTAACTTCTTCAGCCCTC |

## SNP analysis

We searched for tag SNPs of *CMIP* using the Haploview program (http://hapmap.ncbi.nlm.nih.gov/). We found a total of 235 tag SNPs and selected four tag SNPs (rs12925980, rs2287112, rs3751859 and rs77700579) that were associated with some neuropsychiatric disorders in order to determine the associations with SCZ. We searched for minor allele frequencies (MAF) for each SNP across 1,000 genomes. The four SNPs' MAF threshold was set above 0.05 for the Chinese Han population (CHB).

Genomic DNA was extracted from five mL of peripheral blood collected from each subject using a commercial DNA extraction kit (Kangwei Biotech Company, Beijing, China) according to the manufacturer's instructions. SNP genotyping was performed using matrix-assisted laser desorption/ionization time of flight mass spectrometry (MALDI-TOF-MS). The forward and reverse primers for these four SNP amplifications are listed in Table 1.

## Statistical analysis

We compared the demographic variables and allele and genotype distributions between patients and controls using Pearson's chi-square ($\chi^2$) test and Student's *t*-test. Multiple logistic regression was used to test the association between SCZ and alleles or genotypes. IBM SPSS (version 24.0) was used for the statistical analyses mentioned above and R software (version 3.2.3) was used for type I error correction using the false discovery rate (FDR) method. In both case and control groups, we used the goodness of fit $\chi^2$ test to test the Hardy–Weinberg equilibrium (HWE) by online software SNPStats (https://www.snpstats.net/snpstats/start.htm). Haploview 4.2 and SNPStats were then used for linkage disequilibrium (LD) and haplotype analysis. Finally, we used Quanto 1.2.4 software to calculate the statistical power for each SNP according to the MAF (rs12925980: 0.495, rs2287112: 0.175, rs3751859: 0.369 and rs77700579: 0.131). SCZ prevalence was presupposed to be 1% according to previous studies. All tests were two-sided and a $P_{adj}$-value less than 0.05 was considered statistically significant.

# RESULTS

## Demographic characteristics

The case group consisted of 761 SCZ patients (58.2% males, mean age = 34.61 ± 12.02 years) and the control group consisted of 775 healthy people (56.2% males, mean

**Table 2  Test of HWE for case and control groups, all SNPs were in accordance with the HWE in the control group.**

| SNP | Case | | | | Control | | | |
|---|---|---|---|---|---|---|---|---|
| | $H_0$ | He | $\chi^2$ | P | $H_0$ | He | $\chi^2$ | P |
| rs12925980 | 0.483 | 0.486 | 0.026 | 0.871 | 0.488 | 0.481 | 0.186 | 0.666 |
| rs3751859 | 0.449 | 0.441 | 0.278 | 0.598 | 0.415 | 0.435 | 1.739 | 0.187 |
| rs2287112 | 0.212 | 0.241 | 9.975 | 0.002 | 0.228 | 0.224 | 0.298 | 0.585 |
| rs77700579 | 0.173 | 0.173 | 0.006 | 0.937 | 0.188 | 0.191 | 0.199 | 0.655 |

Note:
Ho, observed heterozygosity; He, expected heterozygosity.

**Table 3  Genotypic and allelic distributions of *CMIP* SNPs between SCZ patients and healthy controls in overall subjects.**

| SNPs | Genotype | Case (n) | Control (n) | $\chi^2$ | P | Padj | OR (95 CI) |
|---|---|---|---|---|---|---|---|
| rs77700579 | AA+TA | 733 | 762 | 0.475 | 0.491 | 0.66 | 1 |
| | TT | 7 | 10 | | | | 0.710 [0.268–1.879] |
| | Allele | | | | | | |
| | T | 1338 | 1379 | 0.988 | 0.32 | 0.66 | 1 |
| | A | 142 | 165 | | | | 0.887 [0.700–1.124] |
| rs12925980 | TT+CT | 479 | 499 | 0.193 | 0.66 | 0.66 | 1 |
| | CC | 250 | 273 | | | | 0.953 [0.771–1.179] |
| | Allele | | | | | | |
| | T | 606 | 621 | 0.774 | 0.379 | 0.66 | 1 |
| | C | 852 | 932 | | | | 0.937 [0.810–1.083] |
| rs3751859 | GG+GA | 653 | 685 | 0.387 | 0.534 | 0.66 | 1 |
| | AA | 75 | 87 | | | | 0.901 [0.650–1.250] |
| | Allele | | | | | | |
| | G | 979 | 1050 | 0.201 | 0.654 | 0.66 | 1 |
| | A | 477 | 494 | | | | 1.036 [0.889–1.207] |
| rs2287112 | TT+GT | 682 | 761 | 5.754 | 0.016* | 0.128 | 1 |
| | GG | 24 | 11 | | | | 2.419 [1.175–4.977] |
| | Allele | | | | | | |
| | T | 1214 | 1346 | 0.914 | 0.339 | 0.66 | 1 |
| | G | 198 | 198 | | | | 1.109 [0.897–1.370] |

Notes:
* $P < 0.05$.
$P_{adj}$ represent P corrected by FDR.
*OR*, is abbreviation of Odds ratio; *95%CI* is abbreviation of 95% confidence interval.

age = 34.74 ± 11.41 years). Cases and controls were matched by sex ($\chi^2 = 0.681$, $P = 0.409$) and age ($t = 0.221$, $P = 0.825$). All SNPs were in accordance with the HWE in the control group (Table 2).

## Allele and genotype distribution

rs12925980, rs3751859 and rs77700579 had 98% detection rates and rs2287112 had a 96% detection rate. Table 3 shows the genotypic and allelic distribution of the four SNPs

and the associations with SCZ in the overall sample. The genotypic distribution of rs2287112 was found to be significantly different between SCZ patients and healthy controls ($P = 0.016$), but the difference did not survive the FDR correction adjusted for the multiple comparison ($P_{adj} = 0.128$). The similar distribution differences and associations were observed in the female group (Table 4). The allelic distribution was significantly different between females in the patient and control groups ($P_{adj} = 0.008$). The GG genotype ($OR = 4.445$, 95% CI [1.227–16.105]) and G allele ($OR = 4.445$, 95% CI [1.788–11.046]) were risk factors for SCZ. The statistical power for rs2281112 was 0.949. More details are shown in Table 4. Tables 3 and 4 show the associations based on the recessive genetic model, and the results of other genetic models are listed in Tables S1–S4.

## LD and haplotype analysis

As shown in Fig. 1, the $R^2$ values were 19 across total subjects (A) and male (B) and female subjects (C). LD was not observed across these SNPs according to the criteria ($R^2 > 0.8$). The four SNPs' position relationship in *CMIP* according to the National Center for Biotechnology Information (NCBI, https://www.ncbi.nlm.nih.gov/) gene structure are shown in Fig. 1D. We conducted haplotype association analysis with SCZ across all participants because the LD analysis results were similar between the male and female groups. The haplotype analysis results (Table 4) indicated that the haplotype made of all four SNPs (rs12925980–rs2287112–rs3751859–rs77700579) had a significantly different distribution between SCZ patients and healthy controls ($P_{adj} = 0.018$). Furthermore, we estimated nine common haplotypes with a frequency >1% in detail. The results showed that the haplotype TTGT was significantly associated with SCZ ($OR = 0.42$, 95% CI [0.19–0.94], $P = 0.032$), but when FDR-adjusted the *P*-value was greater than 0.05 (Table 5).

## DISCUSSION

Many studies have investigated the association between the *CMIP* gene and diseases such as mental neuropsychiatric disorder (*Eicher & Gruen, 2015*; *Luo et al., 2017*; *Wang et al., 2015*), cancer (*Juan et al., 2019*), and metabolic disease (*Cao, Wang & Wu, 2018*; *Mo et al., 2018*). In this study, we included 1,536 participants to study the association between four tag SNPs (rs12925980, rs22287112, rs3751859 and rs77700579) of the *CMIP* gene and SCZ. To the best of our knowledge, our study is the first of its kind to explore the correlation between *CMIP* and SCZ in the northeast CHB. We found that one loci (rs2287112) was associated with SCZ in females, indicating that *CMIP* was a potential risk genetic variant for SCZ. A large scale GWAS study conducted by *Gedik (2017)* found that the SNP rs77700579 in *CMIP* was associated with major depressive disorder (MDD), supporting the conclusion that *CMIP* was a potential candidate gene for neuropsychiatric disorders.

Several studies have detected sex-distinct gene polymorphisms with SCZ, including *LTA*, *TNFA*, *IFNGR2* and *PLA2G12A* (*Inoubli et al., 2018*; *Jemli et al., 2017*; *Yang et al., 2016*). *Yu et al. (2013)* found eight genes with differential expression in female and male SCZ patients. Our research group also found a sex-specific SNP of gene *RELN* with SCZ in

**Table 4 Genotypic and allelic distributions of *CMIP* SNPs between SCZ patients and healthy controls stratified by different sex.**

| SNPs | Genotype | Female | | | | | | Male | | | | | |
|---|---|---|---|---|---|---|---|---|---|---|---|---|---|
| | | Case | Control | $\chi^2$ | P | $P_{adj}$ | OR (95% CI) | Case | Control | $\chi^2$ | P | $P_{adj}$ | OR (95% CI) |
| rs3751859 | G/G–G/A | 267 (89.90%) | 302 (89.1%) | 0.111 | 0.784 | 0.824 | 1 | 386 (89.6%) | 383 (88.50%) | 0.27 | 0.578 | 0.845 | 1 |
| | A/A | 30 (10.10%) | 37 (10.90%) | | | | 0.931 [0.559–1.551] | 45 (10.4%) | 50 (11.60%) | | | | 0.886 [0.577–1.358] |
| | Aelle | | | | | | | | | | | | |
| | G | 393 (66%) | 463 (68%) | 0.686 | 0.427 | 0.824 | 1 | 586(68%) | 587 (68%) | 0 | 0.87 | 0.87 | 1 |
| | A | 201 (34%) | 215 (32%) | | | | 1.100 [0.869–1.391] | 276 (32%) | 279 (32%) | | | | 0.983 [0.803–1.204] |
| rs77700579 | A/A-T/A | 305 (99.70%) | 337 (99.4%) | 0.241 | 0.659 | 0.824 | 1 | 428 (98.6%) | 425 (98.20%) | 0.295 | 0.634 | 0.845 | 1 |
| | T/T | 1 (0.30%) | 2 (0.60%) | | | | 1.718 [0.155–19.067] | 6 (1.40%) | 8 (1.80%) | | | | 1.297 [0.445–3.784] |
| | Aelle | | | | | | | | | | | | |
| | A | 556 (91%) | 613 (90%) | 0.072 | 0.824 | 0.824 | 1 | 782 (90%) | 766 (88%) | 1.217 | 0.271 | 0.733 | 1 |
| | T | 56 (9%) | 65 (10%) | | | | 1.044 [0.717–1.520] | 86 (10%) | 100 (12%) | | | | 1.187 [0.875–1.611] |
| rs12925980 | C/C–C/T | 248 (82.9%) | 283 (83.5%) | 0.194 | 0.635 | 0.824 | 1 | 354 (82.3%) | 367 (84.80%) | 0.044 | 0.847 | 0.87 | 1 |
| | T/T | 51 (17.1%) | 56 (16.5%) | | | | 1.083 [0.780–1.504] | 76 (17.7%) | 66 (15.20%) | | | | 1.028 [0.777–1.360] |
| | Aelle | | | | | | | | | | | | |
| | C | 402 (59%) | 348 (58%) | 0.158 | 0.691 | 0.824 | 1 | 726(87%) | 760 (88%) | 0.434 | 0.515 | 0.845 | 1 |
| | T | 276 (41%) | 250 (42%) | | | | 1.051 [0.840–1.314] | 112 (13%) | 108 (12%) | | | | 1.066 [0.879–1.292] |
| rs2287112 | T/T–G/T | 276 (96.2%) | 335 (99.1%) | 6.148 | 0.023 | 0.092 | 1 | 406 (96.9%) | 426 (98.2%) | 1.408 | 0.275 | | 1 |
| | G/G | 11 (3.80%) | 3 (0.90%) | | | | 4.445 [1.227–16.105] | 13 (3.10%) | 8 (1.80%) | | | 0.733 | 1.645 [0.674–4.019] |
| | Aelle | | | | | | | | | | | | |
| | T | 552 (96.20%) | 670 (99.10%) | 12.296 | 0.001 | 0.008* | 1 | 812 (96.90%) | 852 (98.20%) | 2.816 | 0.122 | 0.733 | 1 |
| | G | 22 (3.80%) | 6 (0.90%) | | | | 4.445 [1.788–11.046] | 26 (3.10%) | 16 (1.80%) | | | | 1.645 [0.875–3.094] |

Notes:
* Represent $P_{adj} < 0.05$.
$P_{adj}$ represent $P$ corrected by FDR.
*OR* is abbreviation of Odds ratio, 95% CI is abbreviation of 95% confidence interval.

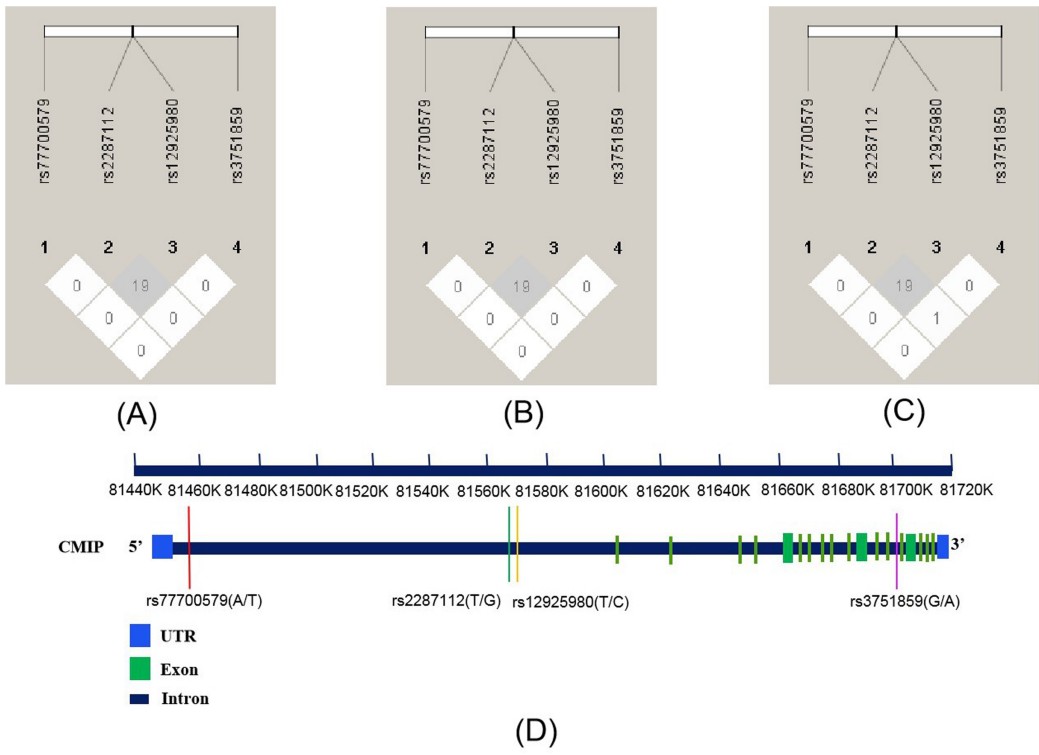

**Figure 1 Linkage disequilibrium (LD) of four SNPs within *CMIP* in different subjects and the location of SNPs on *CMIP* gene structure.** $R^2$ values were used to estimate the LD between pairwise SNPs. (A ) LD of total subjects. (B) LD of male. (C) LD of female. (D) Location of four SNPs on *CMIP* gene.

**Table 5 Association between rs12925980–rs2287112–rs3751859–rs77700579 haplotype and schizophrenia.**

| rs12925980–rs2287711– rs3751859–rs77700579 | Frequency | | | OR (95%CI) | P | $P_{adj}$ |
|---|---|---|---|---|---|---|
| | Total | Control | Case | | | |
| CTGA | 0.3426 | 0.3548 | 0.329 | 1 | | |
| CTAA | 0.1782 | 0.1758 | 0.179 | 1.07 [0.83–1.38] | 0.620 | 0.620 |
| TTGA | 0.1681 | 0.163 | 0.173 | 1.11 [0.85–1.44] | 0.440 | 0.620 |
| TGGA | 0.0945 | 0.0908 | 0.099 | 1.10 [0.80–1.51] | 0.550 | 0.620 |
| TTAA | 0.082 | 0.0769 | 0.09 | 1.26 [0.90–1.74] | 0.170 | 0.453 |
| CTGT | 0.0402 | 0.0357 | 0.047 | 1.33 [0.82–2.17] | 0.250 | 0.500 |
| TGAA | 0.0279 | 0.0309 | 0.025 | 0.83 [0.46–1.51] | 0.540 | 0.620 |
| CTAT | 0.0249 | 0.0307 | 0.019 | 0.60 [0.30–1.22] | 0.160 | 0.453 |
| TTGT | 0.0234 | 0.0317 | 0.013 | 0.42 [0.19–0.94] | 0.032* | 0.272 |

**Notes:**
* $P < 0.05$.
*OR* is abbreviation of Odds ratio, 95% CI is abbreviation of 95% confidence interval.

a previous study (*Bai et al., 2019*). Considering that SCZ's sex-specific molecular phenotype has been observed in previous studies, we first explored the association between *CMIP* and SCZ in all samples and then separately tested the association for the male and female

subgroups. We found that the SNP rs2287112 was significantly associated with SCZ in the whole group and female subgroup with a statistically significant value of 0.05. However, in the whole group the $P$ value did not withstand FDR correction. The association between rs2287112 and SCZ only existed after $P$ value correction in the female group. The association was not observed in the male group, providing more evidence that the molecular phenotype in SCZ is sex-specific. It should be noted that rs2287112 was not in HWE in the SCZ group, which suggested population stratification. The population structure evaluation showed no stratification and the control group conformed to HWE, ruling out the possibility of population admixture. The deviation from HWE may have been caused by the association with the disease that exerted a strong selection on the genome (*Li et al., 2011*).

Additionally, we carried out haplotype analysis to determine the association between the haplotype and SCZ and whether the combination of specific alleles could affect SCZ susceptibility. The TTGT haplotype (rs12925980–rs2287112–rs3751859–rs77700579) correlated with a lower risk of SCZ in our study population, but the association did not survive FDR correction. Similarly, the haplotype consisting of rs12929303–rs2287112–rs12925980 in *CMIP* was associated with developmental dyslexia in a Chinese population (*Wang et al., 2015*), suggesting that the haplotype including rs22287112 may contribute to disease susceptibility. The haplotype analysis further supported that rs2287112 allele G correlated with an increased SCZ risk.

Since this was a cross-sectional study, several limitations should be mentioned. First, this study was limited to interpreting the causal relationship between genetic risk factors and SCZ. Second, we only analyzed four SNPs in this study and may have missed some other loci associated with SCZ. Additionally, owing to the failure of demographic characteristic and in-depth clinical trait collection, we were not able to analyze the association of these SNPs with different SCZ clinical features. We were also limited to interaction analysis between genes and environment. Further studies that incorporate a large-scale sample size with more demographic characteristic information are warranted to further substantiate the association between *CMIP* gene polymorphism and SCZ susceptibility.

## CONCLUSION

This study presented evidence that a *CMIP* variant is associated with SCZ susceptibility in northeast Han Chinese women. Considering the limitations of our work, additional functional genomics studies are required to further explain the role of SCZ-associated *CMIP* variants.

## ACKNOWLEDGEMENTS

We are grateful to all DNA sample donors and the research assistants who helped with sample collection.

### Funding

This work was supported by the National Natural Science Foundation of China (No. 81673253) and the Youth Development Fund from First Hospital of Jilin University (No.

JDYY11202021). The funders had no role in study design, data collection and analysis, decision to publish, or preparation of the manuscript.

### Grant Disclosures
The following grant information was disclosed by the authors:
National Natural Science Foundation of China: 81673253.
Youth Development Fund from First Hospital of Jilin University: JDYY11202021.

### Competing Interests
The authors declare that they have no competing interests.

### Author Contributions
- Yingli Fu conceived and designed the experiments, performed the experiments, analyzed the data, prepared figures and/or tables, authored or reviewed drafts of the paper, and approved the final draft.
- Xiaojun Ren performed the experiments, authored or reviewed drafts of the paper, and approved the final draft.
- Wei Bai analyzed the data, prepared figures and/or tables, and approved the final draft.
- Qiong Yu conceived and designed the experiments, authored or reviewed drafts of the paper, and approved the final draft.
- Yaoyao Sun performed the experiments, prepared figures and/or tables, and approved the final draft.
- Yaqin Yu conceived and designed the experiments, authored or reviewed drafts of the paper, and approved the final draft.
- Na Zhou analyzed the data, authored or reviewed drafts of the paper, and approved the final draft.

### Human Ethics
The following information was supplied relating to ethical approvals (*i.e.*, approving body and any reference numbers):
The Ethics Committee of Jilin University, China approval to carry out the study within its facilities (Ethical Application Ref: 2014-05-01).

### Data Availability
The data and a codebook for the data are available as Supplemental Files. The results of this article were all calculated from this raw data.

### Supplemental Information
Supplemental information for this article can be found online at http://dx.doi.org/10.7717/peerj.11907#supplemental-information.

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
