# Peer review of "Association between C-Maf-inducing protein gene rs2287112 polymorphism and schizophrenia"

_PeerJ, doi:10.7717/peerj.11907_

## Round 0.1 · original submission · Major Revisions

Thank you for submitting your interesting work to us. Although the study has merits, our reviewers have major concerns regarding its writing quality and methodology. Please address these issues accordingly.

Reviewer 1 ·

Basic reporting

None

Experimental design

None

Validity of the findings

None

Additional comments

This paper conducted a genetic study that have evaluate the genetic association between CMIP gene and schizophrenia in a Han Chinese population. Although this is a positive study, which may provide valuable information, there are several major problems:
(1) Please provide the full name of CMIP in the title and abstract
(2) More background information about the relationship between CMIP gene variant and schizophrenia, latest progress and related mechanisms should be provided to tell readers why you exert the study.
(3) Please add related hypotheses in the introduction.
(4) Please provide a table for primer information.
(5) Authors should provide more demographic characteristics in the included subjects in order to further explore the interaction between gene and environment.
(6) Sex difference should provide the completed findings, such as alleles and genotypes distribution, model analysis, linkage disequilibrium and haplotype analysis.
(7) Please re-check table 3.
(8) Please provide a figure about linkage disequilibrium in total population.

Reviewer 2 ·

Basic reporting

In this research, Fu et al., conducted association analyses between 4 SNPs in CMIP with schizophrenia and found the allele frequency of rs287112 is significantly different between schizophrenia patients and healthy controls in female Han Chinese population. Further analyses showed that haplotype TTGT of rs12925980-rs2287112-rs3751859-rs77700579 had a lower risk for schizophrenia when compared to CTGA haplotypes. In all, the manuscript is clearly written except for some minor problems. However, whether such an association report is qualified to published in PeerJ needs to be considered. Here are my detailed comments:
1. As this study is an association analyses, there is no one reference about the genome wide association study (GWAS) of schizophrenia. In recent years, many GWASs with more than 100,000 samples were reported, like [Pardinas et al., 2018, Nat Genet; Lam et al., 2019, Nat Genet; PGC, 2020, medRxiv]. Specially, the authors should read the Lam paper, which reported the largest study to date of East Asian participants (22,778 schizophrenia cases and 35,362 controls), and query whether the 4 SNPs (rs12925980, rs2287112, rs3751859 and rs77700579) were significantly associated with schizophrenia.
2. If the rs287112 is related to schizophrenia, the authors should verify this result in large samples.
3. About figure 1, there is no need to show D’, R2 is enough to elucidate the LD. In addition, the authors can add the structure information of CMIP gene above the figure to show the position relationship of 4 SNPs and CMIP.

About the writing:
1. Line 37-38, “...in a Han Chinese female population” to “...in the Han Chinese female population”? Please consider it clearly.
2. Line 89-90, “the threshold for MAF of all these four SNPs was set 90 greater than 0.05 in Chinese Han population (CHB).”, here, “set” may be not suitable.
3. Line 96-99, a space in the primer sequences. Please check it carefully.
4. Line 159, “...in CMIP was to be associated with...” to “...in CMIP was associated with...”

Experimental design

no comment

Validity of the findings

no comment

Additional comments

no comment

---

## Round 0.2 · Major Revisions

English language is still problematic. Please have the paper strictly polished by native speakers to ensure its accuracy.

Reviewer 1 ·

Basic reporting

None

Experimental design

None

Validity of the findings

None

Additional comments

Thank you for your revision. But I still have some minor issues as follow:

1 please recheck the units and names to ensure the accuracy.
2. Some abbreviations such as GWAS, gene names should be presented full name when they first used.
3. Line 77. "...a Han Chinese population" to "the Han Chinese population".
4. Genetic models will give us more information. So I suggest authors to perform related analyses.
5.Line 192. "presented evidence" to "presented an evidence".
6 .Table 3. Please use “lower n” rather than "upper N"
7.Table 3 ,4 and 5. Please add the footnote about OR and CI.
8. Continuous Table 4 is same as Table 4. Please check them.
9. Figure 1. Please provide legend about A .B .C .D

Reviewer 2 ·

Basic reporting

no comment

Experimental design

no comment

Validity of the findings

no comment

Additional comments

The authors have addressed all of my concerns.

---

## Round 0.3 · accepted · Accept

I am satisfied with this revised version of the paper.

Reviewer 1 ·

Basic reporting

None

Experimental design

None

Validity of the findings

None

Additional comments

None